# Deep Brain Stimulation for Gilles de la Tourette Syndrome: Toward Limbic Targets

**DOI:** 10.3390/brainsci10050301

**Published:** 2020-05-15

**Authors:** Domenico Servello, Tommaso Francesco Galbiati, Roberta Balestrino, Guglielmo Iess, Edvin Zekaj, Sara De Michele, Mauro Porta

**Affiliations:** 1Neurosurgical Department, IRCCS Istituto Ortopedico Galeazzi, 20100 Milan, Lombardia, Italy; servello@libero.it (D.S.); guglielmoiess@gmail.com (G.I.); ezekaj@yahoo.com (E.Z.); 2Department of Neuroscience “Rita Levi Montalcini”, University of Turin, 10100 Torino, Italy; roberta.balestrin@edu.unito.it; 3Tourette’s Syndrome and Movement Disorders Center, IRCCS Istituto Ortopedico Galeazzi, 20100 Milan, Lombardia, Italy; sarademichele1@gmail.com (S.D.M.); portamilano@libero.it (M.P.)

**Keywords:** Deep Brain Stimulation, antero-medial Globus Pallidus internus, ventralis oralis/centromedian-parascicular nucleus of the thalamus, Tourette Syndrome, obsessive compulsive disorder

## Abstract

Gilles de la Tourette syndrome (GTS) is a complex neurodevelopmental disorder characterized by tics and, frequently, psychiatric and behavioral comorbidities. Above all, obsessive compulsive disorder/behavior (OCD/OCB) influences the clinical picture and has a severe impact on quality of life, eventually more than the tics themselves. Deep brain stimulation (DBS) is an effective therapy in selected, refractory cases. Clinical response to DBS may vary according to the clinical picture, comorbidities, and to the anatomical target. This retrospective study compares the results obtained from DBS in the ventralis oralis/centromedian-parascicular nucleus of the thalamus (Voi-Cm/Pf) (41 patients) and antero-medial Globus Pallidus internus (am-GPi) (14 patients), evaluating clinical response over time by means of Yale Global Tic Severity Scale (YGTSS) and Yale–Brown Obsessive-Compulsive Scale (YBOCS) scores over a period of 48 months. A significant and stable improvement in the YGTSS and YBOCS has been obtained in both groups (*p* < 0.001). There was a significant difference in YBOCS improvement over time between the am-GPi group and the Voi-Cm/Pf group, indicating a better and faster control of OCD/OCB symptoms in the former group. The ratio of hardware removal was 23% and limited to 13 patients in the Voi-Cm/Pf group. These results confirm that DBS is an effective therapy in treating GTS and suggest that the am-GPi might be superior to Voi-Cm/Pf in alleviating comorbid OCD/OCB.

## 1. Introduction

Gilles de la Tourette syndrome is a complex, relatively common neurodevelopmental disorder characterized by tics, which are defined as sudden tonic, clonic, or dystonic involuntary movement and sounds [1]. The basal ganglia and the cortico–striatal–thalamo–cortical circuit seem to play a relevant role in the pathophysiology of GTS [2,3,4]: they have been identified together with the frontal cortex and other limbic structures as a possible “tic-generating network” [5]. The prevalence of GTS is approximately 1% and the onset usually occurs between 6–8 years of age. The severity of tics typically follows a waxing and waning course, reaching a climax at 10–12 years of age, followed by a significant improvement during adolescence and early adulthood, although this does not happen in all patients. Some patients develop what has been defined by some authors as “malignant GTS,” a form of GTS characterized by self-injurious behaviors that can result in hospitalizations and temporary (or even permanent) disabilities [6]. Comorbidities, especially psychiatric ones, are an integral part of the clinical picture and contribute to its complexity: GTS is frequently associated with psychopathologies and behavioral aspects, such as obsessive compulsive disorder/behavior (OCD/OCB), attention deficit hyperactivity disorder (ADHD), self-injurious behaviors (SIBs), depression, anxiety, and personality disorders; less common aspects are conduct disorder, oppositional defiant disorder, learning difficulties, and aggressiveness [7]. According to the presence of comorbidities, five subtypes of GTS can be recognized [8]. Specifically, obsessive-compulsive tic disorder (OCTD) is a GTS subtype characterized by OCD/OCB as principal feature, together with tics. We have previously demonstrated that comorbidities, in particular OCD/OCB, can cause stigma and isolation, reduced work and school performance and—in conclusion—a more severe impact on quality of life than the tics themselves; therefore, when making therapeutic decisions, we recommend considering comorbidities and social/working/school impairment besides of the mere severity of tics [9]. Treatments for GTS involve psychological, behavioral and social interventions, as well as pharmacological therapy; mainly, but not exclusively, dopamine antagonists or depleters, antiepileptic drugs, α2 adrenergic agonists, and botulinum toxin injections. Deep Brain Stimulation (DBS) is an option in GTS patients who are refractory to standard treatments and who experience a severe impact on QoL as well as possible complications [10].

Studies on DBS in GTS are scarce and difficult to compare, as studies mostly report few or even only single cases. To date, at least seven different brain areas have been targeted; among them, thalamic stimulation is the most described and has been used in more than 50% of all the patients reported [11,12].

The aim of this study is to compare the efficacy of the ventro-oralis-internus centromedian parafascicular thalamus and antero-medial limbic part of the Globus Pallidus internus (GPi) as DBS targets for GTS.

## 2. Methods

### 2.1. Patients Cohort and Surgical Procedure

In this retrospective, longitudinal observational study we reviewed medical chart data and health information of all the 63-consecutive treatment-refractory GTS and OCDT patients treated with bilateral DBS at our institution between 2004 and 2019. Patients implanted in the nucleus accumbens/anterior limb of the internal capsule (NAc/ALIC), multiple bilateral target, and those who were re-implanted after a removal were excluded from the analyses.

Patients were selected for DBS therapy following published recommendations [10]. The implantation of electrodes for bilateral DBS was guided by individual targeting on stereotactic T1 or proton-density magnetic resonance imaging (MRI), visualizing the individual thalamic or pallidal target. Thalamic target identified on the following stereotactic coordinates: 5 mm lateral to the anterior- and posterior commissure line (AC-PC), 2 mm posterior to mid-commissural point (MCP), and at the AC-PC plane. This target is different than the one used by Vandewalle and colleagues, as it is located 2 mm more anteriorly [13,14] (Figure 1). The area of electrode positioning in the antero-medial Globus Pallidus internus (am-GPi) was based on direct target visualization on MRI (Figure 2). All patients were implanted bilaterally with quadripolar DBS leads (Medtronic 3389 and 3391). In order to precisely localize the DBS electrodes, computed tomography (CT) or MRI was performed in every patient, with a voxel size less than 3.0 × 3.0 × 3.0 mm. The primary outcome was the Yale Global Tic Severity Scale total score. The secondary outcome was the Yale–Brown Obsessive-Compulsive Scale score.

### 2.2. Statistical Analisis

Estimates of monthly symptoms decrease within groups were calculated using linear mixed models (with time as a fixed factor). We adopted mixed-effect regression models with random intercepts, with time and target playing the role of covariates. The hypothesis under test is a difference in the rate of change between the two slopes (as measured by the YGTSS and the YBOCS) of the am-GPi and Voi-Cm/Pf cohorts.

A series of post hoc comparisons were then carried out between baseline and follow-ups time points within each group. Adjustment for multiple comparisons was made utilizing the Bonferroni method.

The results were analyzed using SPSS (IBM Corp. 2020 Release, IBM SPSS Statistics for MacOs, Version 26.0. Armonk, NY, USA). All *p*-values reported are two-tailed, and a *p* < 0.05 was considered statistically significant. Descriptive statistics (mean, standard deviation, and range) were used for continuous variables and frequency for categorical data.

### 2.3. Ethics

Written informed consent was obtained from all the participants. Data were analysed anonymously; the local ethical committee (IRCCS Istituto Ortopedico Galeazzi, Milan, Lombardia, Italy) approved the study. All performed procedures were in compliance with the Declaration of Helsinki and its later amendments or comparable ethical standards and with the ethical standards of the Lund University research committee (2014/403). 

## 3. Results

Fifty-five patients were included in the analysis; among them, 41 patients were implanted in the Voi-Cm/Pf nucleus and 14 in am-GPi. There were 31 males and 10 females in the thalamus group and 11 males and three females in the am-GPi group. There was no significant difference in the mean age at surgery (31 years old in the thalamus group, 28 in the am-GPi group) (*p* = 0.20). Baseline and follow-up scores in evaluation scales are reported in Table 1 and Table 2.

As mentioned above, comparisons and estimates were made with mixed linear model regressions. Average decrease between two measurements of YGTSS in Voi-Cm/Pf and am-GPi was −9.20 and −12.73 respectively, while that of YBOCS in Voi-Cm/Pf and am-GPi was −1.51 and −4.45 respectively.

Comparison between baseline and follow-ups within the same groups showed significant improvement of both YGTSS and YBOCS between t0 and t1, t0-t2, t0-t3, t0-t4 in the thalamus group and in the GPi group (*p* < 0.001, except for t0–t3 of YBOCS am-GPi group with a *p* value < 0.037, probably due to missing data during patient follow-ups).

Univariate tests evaluating the effect of time on the decrease of symptoms on linearly independent pairwise comparisons among the estimated marginal means was significant (*p* < 0.001) in both Voi-Cm/Pf (F(4, 73.20) = 46.56) and am-GPi groups (F(4, 20.51) = 56.62) of YGTSS and was also significant (*p* < 0.001) in Voi-Cm/Pf (F(4, 77.52) = 8.28) and am-GPi (F(4, 16.43) = 21.38) of YBOCS.

During the 48 months follow-up period after surgical treatment, the difference in the rate of decrease in symptoms severity between the two groups, as measured by the YGTSS, was found not to be statistically significant. We found a different result for the YBOCS measure, which exhibited a significantly faster decrease in the am-GPi group than in the Voi-Cm/Pf group (difference in slope −2.8; *p* < 0.001; F (1185.34) = 18.89), convincingly indicating the benefit of am-GPi in reducing obsessive-compulsive behavior ( Figure 3 and Figure 4).

We further analyzed YGTSS and YBOCS with multiple comparison post-hoc tests (corrected with Bonferroni method) calculating differences between the two groups at each of the five time points. While in t1, t2, t3, and t4 no significant differences were found, t0 yielded a *p* = 0.13 (YGTSS) and *p* < 0.001 (YBOCS), indicating a higher symptom severity score at baseline in the am-GPi group.

13 patients in the thalamus group underwent electrode removal due to side effects (skin erosion and infection, *n* = 8; poor compliance, *n* = 3; resolution of tics, *n* = 2; mean time from implantation to explantation 4.61 years), no patient in the GPi group underwent removal.

## 4. Discussion

The first target for DBS used in GTS has been, more than 20 years ago, the ventralis oralis internus nucleus of the thalamus and the medial part of the centromedian/(Voi/Cm) nucleus of the thalamus [15]; the selection of this target was initially driven by the positive outcome obtained with thalamotomy in treating GTS reported by Hassler and Dieckmann [16]. Since then, several different targets have been used [12]. These targets include: Voi/CM-Pf of the thalamus, the ventro-postero-lateral motor part of the GPi, the am-Gpi, and the NAc/ALIC.

Currently, targets are usually selected based on clinical presentation, based on the evidence emerging from the available literature: indeed, the Cm/Pf is generally preferred for patients with predominant tics and milder comorbidities, whereas the NAc, the ventral striatum, ALIC, and the am-GPi are preferred in patients with a predominant comorbidity burden [11,12,17].

An increasing body of evidence supports the choice of the GPi: in a recent review, the anterior GPi led to the greatest improvement (55.3%) in the YGTSS compared with other targets [18]. In the recently published data from a multicenter registry including 185 GTS patients treated with DBS, the YGTSS improvement at one year of follow-up was 50.5% in the anterior GPi group, 46.3% in the centro-median thalamic region group, and 27.7% in the posterior GPi group [19]. More recently, an analysis of 110 patients from the International TS-DBS Database and Registry found no differences in YGTSS and YBOCS score among time between patients implanted in Voi-Cm/Pf and am-GPi [12]. Nonetheless, the strict interconnections of am-GPi with prefrontal circuits might explain the efficacy of this target in alleviating symptoms of OCD, as previously reported [20].

Our experience and available evidence highlight OCD/OCB as the most debilitating and disturbing aspect of GTS, especially in OCDT patients; therefore, we aimed to prioritize the treatment of this feature and tried to select DBS targets accordingly.

In fact, we have progressively located the Voi-CM/Pf target 2 mm more anteriorly than previous reports, aiming for a better stimulation of the associative-limbic connections, to target both motor and behavioral manifestations of GTS [13,14,21]. We have preferred the NAc/ALIC to treat patients with comorbidities, especially OCD/OCB: therefore, in patients suffering from debilitating OCD/OCB despite a previous thalamic DBS or as a baseline condition the NAc has been used as an add-on rescue or as concomitant (to thalamus) DBS target [18]. However, we have observed a better efficacy of the am-GPi in one patient (not included in this study, as aforementioned in methods) who underwent bilateral Voi-Cm/Pf and NAc-ALIC DBS: the latter intervention led to suboptimal OCD symptoms control; the patients continued to experience severe social impairment, and developed compulsive scratching of the surgical wound, which caused an infection and led to hardware removal. Due to the unsatisfactory results, the patient then underwent bilateral am-GPi DBS, this time with satisfactory results on both tics and OCD/OCB. Previous experience and emerging evidence increased our interest in the am-GPi as an optimal target for GTS, including OCDT, and in this study, we therefore compared the results obtained in GTS/OCDT patients who underwent thalamic and GPi DBS. While the two groups did not differ with regards to motor tics before surgical treatment, am-GPi group patients presented a significant difference in YBOCS at baseline; this could be due to the lack of randomization and represents a limitation of this study. 

However, while the am-GPi DBS was as effective as the Voi-Cm/Pf in reducing motor tics, it was more effective in controlling OCD/OCB comorbidities; moreover, the response appeared to be faster than in the thalamus group. Indeed, when we confronted the two rates of YBOCS symptoms decrease, am-GPi was clearly superior. Furthermore, patients treated with am-GPi DBS showed even a lower risk of developing hardware-related adverse events compared to those who underwent Voi-Cm/Pf DBS, as emerges by the different number of patients who underwent removal in the two groups. As aforementioned, the higher ratio of leads removal in thalamus group could be explained by a reduced efficacy in alleviating OCD symptoms and the high prevalence of scratching over surgical wounds (a typical behavior in patients with OCD/OCB) we have observed in our patients, as compared to patients in the am-GPi, who seem to not have this symptom, possibly due to the greater efficacy of this target in reducing OCD/OCB.

The strengths of this study include the number of patients (to our knowledge, the largest cohort of DBS-GTS patients worldwide), the length of the follow up and the completeness of our data. Moreover, all the patients have been treated by the same neurologist (MP) and neurosurgeon (DS), therefore avoiding differences in the management of the patients. The limitations of this study include the different number of patients who underwent thalamus vs. am-GPi DBS, the significant difference in YBOCS at baseline in the two groups, the different distribution in time, and the lack of randomization. 

## 5. Conclusions

Although in most cases GTS is “benign” and responds to therapy or spontaneously resolves in adulthood, in some cases, it might have more severe characteristic, causing a severe impact on patients’ lives [22]. Despite the severity of this condition, therapeutic options are limited. Our study confirms the efficacy of DBS as a treatment for select patients with GTS. In our cohort, we observed a significant improvement in both tics and comorbidities, reduction of pharmacological therapy, and improvement of quality of life. We have observed a better outcome in patients who underwent am-GPi DBS: despite worse baseline comorbidities, these patients showed a satisfactory response to DBS, especially in OCD/OCB comorbidities, and showed a faster and better response. The benefits in both groups were sustained over a long time, but patients in the am-GPi experienced fewer complications and none of the patients in this group underwent electrode removal (vs. 31% in the thalamus group). Therefore, we believe that in selected GTS patients, especially those with OCDT, the am-GPi is an optimal target for DBS. Nevertheless, larger, multicentre studies are needed to effectively compare different DBS targets in GTS.

## Figures and Tables

**Figure 1 brainsci-10-00301-f001:**
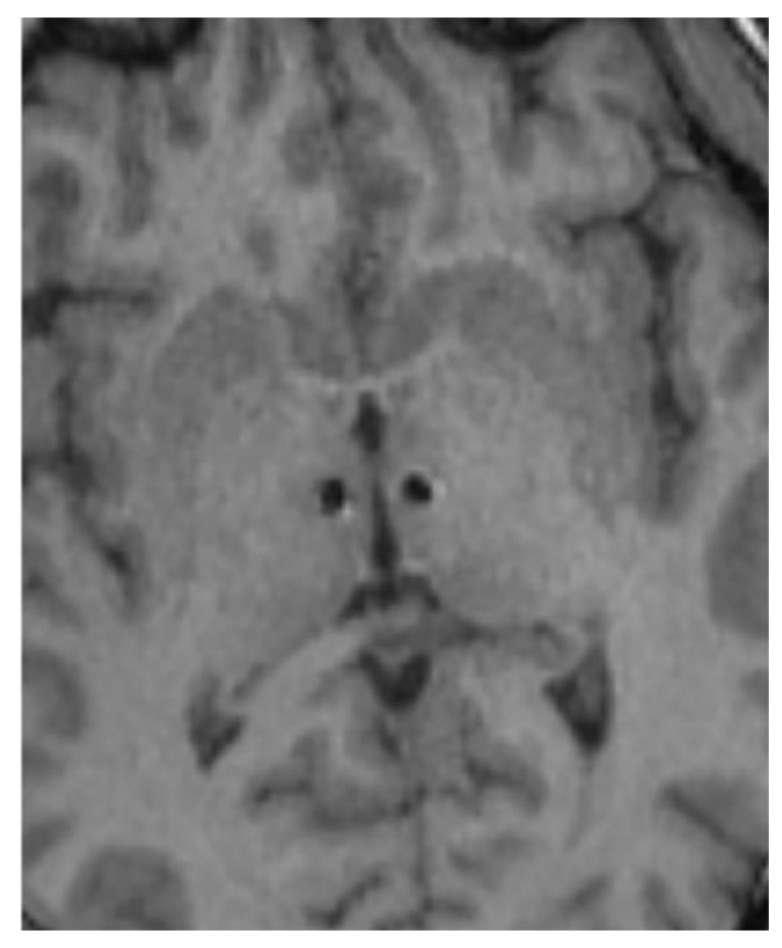
Axial T1-weighted magnetic resonance imaging (MRI) showing electrodes in the ventralis oralis/centromedian-parascicular nucleus of the thalamus (Voi-Cm/Pf) (1.5 Tesla).

**Figure 2 brainsci-10-00301-f002:**
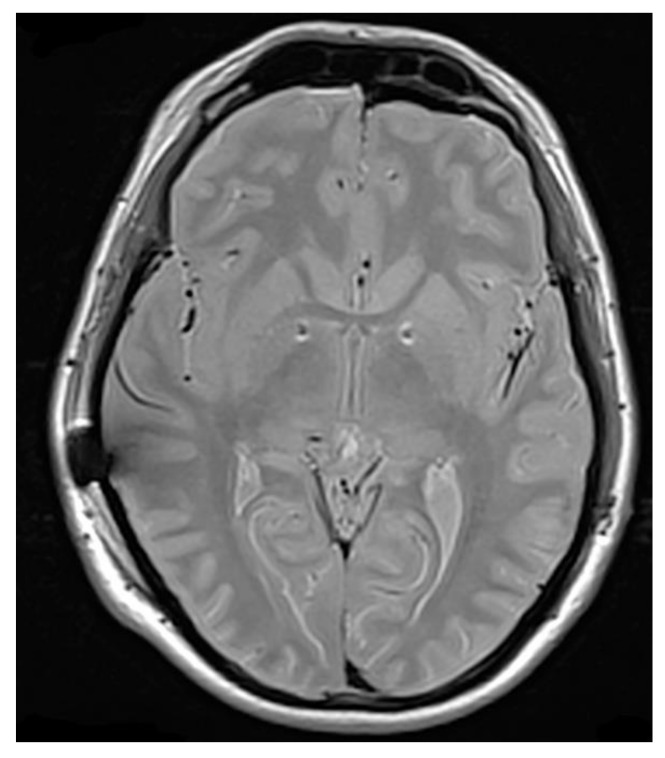
Axial T2-weighted MRI sequences showing electrodes in the anteromedial part of globus pallidus internus (1.5 Tesla).

**Figure 3 brainsci-10-00301-f003:**
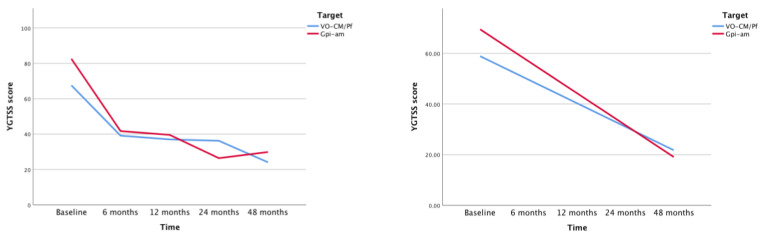
Changes in YGTSS during time. The graph on the left shows difference between slopes of antero-medial Globus Pallidus internus (am-GPi) (red) and Voi-Cm/Pf (blue). The graph on the right represents its linear regression using mixed models.

**Figure 4 brainsci-10-00301-f004:**
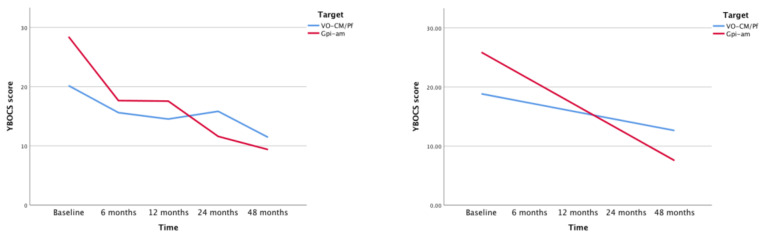
Changes in YBOCS during time. The graph on the left shows difference between slopes of am-GPi (red) and Voi-Cm/Pf (blue). The graph on the right represents its linear regression using mixed models.

**Table 1 brainsci-10-00301-t001:** Yale Global Tic Severity Scale (YGTSS) and Yale–Brown Obsessive-Compulsive Scale (YBOCS) scores at t0 (baseline) and follow-ups (t1 = 6 months, t2 = 12 months, t3 = 24 months, t4 = 48 months) in the ventralis oralis/centromedian-parascicular nucleus of the thalamus.

		YGTSS t0	YGTSS t1	YGTSS t2	YGTSS t3	YGTSS t4	YBOCS t0	YBOCS t1	YBOCS t2	YBOCS t3	YBOCS t4
N	Valid	41	41	36	30	29	41	41	36	30	29
	Missing	0	0	5	11	12	0	0	5	11	12
Mean		67.56	39.12	37.00	36.23	24.07	20.17	15.61	14.53	15.83	11.45
Median		69.00	40.00	38.00	36.00	20.00	21.00	16.00	14.50	16.00	10.00
Std. Deviation		19.166	14.755	14.672	13.392	10.166	9.420	7.697	7.565	9.135	8.716
Range		85	63	65	57	45	38	31	28	40	30
Minimum		12	7	5	14	5	0	0	0	0	0
Maximum		97	70	70	71	50	38	31	28	40	30

**Table 2 brainsci-10-00301-t002:** YBOCS and YBOCS scores at t0 (baseline) and follow-ups (t1 = 6 months, t2 = 12 months, t3 = 24 months, t4 = 48 months) in the anterior-medial Globus Pallidus internus (GPi) group.

		YGTSS t0	YGTSS t1	YGTSS t2	YGTSS t3	YGTSS t4	YBOCS t0	YBOCS t1	YBOCS t2	YBOCS t3	YBOCS t4
N	Valid	14	14	9	5	8	14	14	9	5	8
	Missing	0	0	5	9	6	0	0	5	9	6
Mean		82.57	41.71	39.56	26.40	29.88	28.43	17.64	17.56	11.60	9.38
Median		83.50	42.50	36.00	28.00	30.00	29.00	18.50	20.00	10.00	9.00
Std. Deviation		10.938	16.845	15.067	7.829	15.496	6.665	7.132	6.247	3.209	3.701
Range		40	61	49	21	50	23	24	16	7	10
Minimum		55	18	21	14	10	13	8	8	8	5
Maximum		95	79	70	35	60	36	32	24	15	15

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
