# Peer review of "Deep Brain Stimulation for Gilles de la Tourette Syndrome: Toward Limbic Targets"

_brainsci, 2020, doi:10.3390/brainsci10050301_

Round 1

Reviewer 1 Report

In this manuscript, Servello and the authors analyzed the effects of thalamic versus am-GPi DBS in Tourette syndrome in a retrospective cohort. The design is straightforward and the results are clear.

  1. In line 140, patients underwent thalamic DBS may also receive add-on rescue with NAc DBS. While the main reason to perform NAc DBS is to treat OCS/OCB, it may also improve tics symptoms which serve as the primary outcome. Please discuss or provide relevant data about patients with/without or before/after NAc DBS.
  2. In line 142, please correct “am-GPI” to am-GPi for internal consistency.

Author Response

  1. In line 140, patients underwent thalamic DBS may also receive add-on rescue with NAc DBS. While the main reason to perform NAc DBS is to treat OCS/OCB, it may also improve tics symptoms which serve as the primary outcome. Please discuss or provide relevant data about patients with/without or before/after NAc DBS.

We thank the reviewer for his/her interesting comment. We have added a brief explanation of the indication for choosing the NAc as a target, as highlighted by the reviewer, and we have reported our own experience on a patient who who underwent bilateral Voi-Cm/Pf and NAc-ALIC and then , due to suboptimal control of both the OCD/OCB and tics, undervent bilateral am-GPi DBS with satisfactory results. However in our case series we have only 9 patients in wich NAc-ALIC has been chosen as primary, concomitant or as add-on rescue target. In these cases the decision to select NAc-ALIC was based more on presence/absence of OCB/OCD than the tics itself.

  1. In line 142, please correct “am-GPI” to am-GPi for internal consistency.

We have modified it according to your comment.

Reviewer 2 Report

I’m facing a study that reports on a retrospective efficacy analysis using Cm/PF and GPI as targets for DBS interventions for patients suffering from Ts. Albeit the study presents a comparably high sample size and a long observational time-window, it lacks not only statistical rigour (main limitation) but also is characterized by insufficient representation of the data (such as the missing of an entire figure, wrong figure labels and references).

More specifically:

Statistical rigour: The authors performed multiple t-Tests without outlining how they corrected for multiple-comparisons (I fear they haven’t; besides that, the tests they have selected are not appropriate for these type of data (see: unbalanced design, potential multicollinearity, to mention a few)). They also mention of having performed an ANOVA without specifying it (I assume, or hope, they have performed a rmANOVA)! They should inform themselves about best practices for statistical data analysis and reporting (among many caveats, e.g. the F- values, df etc are entirely lacking; no mentioning of pairwise comparisons and interactions, no table of descriptive analysis of the sample, among many other issues). Based on this, I question the validity of the results that are presented.

Scientific rigour: Figure 2 is entirely missing, Figure 4, which, strangely enough, is positioned within the discussion-section, is incorrectly referred to, many typos e.g. in figure captions etc.

Language: There are many inaccurate (and sometimes wrong) statements with regard to standard statistical reporting and representation of results. In addition, the reviewer thinks this work would require extensive editing of English language and style.

Author Response

  1. Statistical rigour: The authors performed multiple t-Tests without outlining how they corrected for multiple-comparisons (I fear they haven’t; besides that, the tests they have selected are not appropriate for these type of data (see: unbalanced design, potential multicollinearity, to mention a few)). They also mention of having performed an ANOVA without specifying it (I assume, or hope, they have performed a rmANOVA)! They should inform themselves about best practices for statistical data analysis and reporting (among many caveats, e.g. the F- values, df etc are entirely lacking; no mentioning of pairwise comparisons and interactions, no table of descriptive analysis of the sample, among many other issues). Based on this, I question the validity of the results that are presented.                                                                               

Thank your for your interesting comments. In light of your precious observations, we decided to select mixed effect models to analyse our data because they are robust to unbalanced design and possible bias due to loss of data during follow-ups (that hampered the possibility of using repeated measure ANOVA). 

See line 306 for the description of statistical analysis: Estimates of monthly symptoms decrease within groups were calculated considering time as a fixed factor. Moreover, to test the difference concerning the rate of change between the two slopes (as measured by the YGTSS and the YBOCS) of the am-GPi and Voi-Cm/Pf cohorts, we generated mixed-effect regression model with random intercepts and time and target playing the role of covariates. A series of post hoc comparisons were then carried out between baseline and follow-ups time points within each group. Adjustment for multiple comparisons was made utilizing the Bonferroni method. Since we were only interested in evaluating clinical improvement of each scale separately, accounting for possible multicollinearity effects was not necessary. The results were analyzed using SPSS (IBM Corp. 2020 Release, IBM SPSS Statistics for MacOs, Version 26.0.).

According to your suggestions, we have modified the results section:

Please see line 375: Average decrease between two measurements of YGTSS in Voi-Cm/Pf and amGPi was -9.20 and -12.73 respectively, while that of YBOCS in Voi-Cm/Pf and amGPi -1.51 and -4,45 respectively.

Comparison between baseline and follow-ups within the same groups showed significant improvement of both YGTSS and YBOCS between t0 and t1, t0-t2, t0-t3, t0-t4 in the thalamus group and in the GPi group (p<0.001, except for t0-t3 of YBOCS amGPi group with a p value < 0,037, probably due to missing data during patient follow-ups).

Univariate tests with mixed effect model evaluating the effect of time on the decrease of symptoms on linearly independent pairwise comparisons among the estimated marginal means was significant (p<0.001) in both Voi/Cm-Pf [F(4, 73.20)=46,56] and am-GPi groups [F(4, 20.51)=56.62] of YGTSS and was also significant (p<0.001) in Voi-Cm/Pf [F(4, 77.52)=8.28] and amGPi [F(4, 16.43)=21.38] of YBOCS.

During the 48 months follow-up period after surgical treatment, the difference in the rate of decrease in symptoms severity between the two groups, as measured by the YGTSS, was found not to be statistically significant. We found a different result for the YBOCS measure, which exhibited a significantly faster decrease in the am-GPi group than in the Voi-Cm/Pf group [difference in slope -2.8; p<0.001; F(1,185.34)=18,89], convincingly indicating the benefit of am-GPi in reducing obesessive-compulsive behaviour. We further analysed YGTSS and YBOCS with post-hoc tests calculating differences between the two groups at each of the 5 time points. While in t1, t2, t3, t4 no significant differences were found, t0 yielded a p<0.13 (YGTSS) and p<0.001 (YBOCS) indicating a higher symptom severity score at baseline in the amGPi group

  1. Scientific rigour: Figure 2 is entirely missing, Figure 4, which, strangely enough, is positioned within the discussion-section, is incorrectly referred to, many typos e.g. in figure captions etc.

We have modified the figures according to your suggestion: figure 2 has been divided from figure 1; figure 3 and 4 has been added in results section. All the figure captions has been corrected.

  1. Language: There are many inaccurate (and sometimes wrong) statements with regard to standard statistical reporting and representation of results.

Statistical reporting has been expanded as aforementioned. We have provided all the statistical analyses.

  1. In addition, the reviewer thinks this work would require extensive editing of English language and style.

Thank you for your helpful comments. We have reviewed our manuscript with the help of an English native speaker and implemented several corrections the text in aim to improve the language.

Round 2

Reviewer 2 Report

I’m facing a revision of the study reporting on a retrospective efficacy analysis using Voi-Cm/PF and am-GPI as targets for DBS interventions for patients suffering from Ts. The authors have invested a great amount of efforts to improve the analysis as well as English language and style and present a more balanced manuscript. I think the main outcome demonstrating the efficacy of DBS to treat refractory GTS patients in both groups is important. There are, however, still one main issue and two secondary issues that need further clarification at this timepoint. I therefore select major revision because I want to reevaluate the paper after the authors have provided information on these issues.

Major concerns:

The authors refrain from outlining the sample size of patients at each different follow-up timepoint. According to Figure 5 (line 245), the reviewer assumes 5 patients at the maximum that completed the long-term follow up in the am-GPi study arm. If that were the case, the low sample size would not only impact on the statistical analysis but also on the confidence with which one can draw conclusions. Underpowered studies are a XX. Therefore, please provide the exact sample size for each study-group (Voi-Cm/PF and am-GPi) and at each follow-up. Please provide this important bit of information (lines 138 and following) so that the reviewer can assess the validity of the results and make sure, that the study is not underpowered (causing the danger of inflating effects where there are none). Otherwise, the authors would need to emphasize that parts of their analysis were performed using a small sample, outlining why this was justifiable from a statistics point of view and highlight that caution is warranted when interpreting these results. Maybe they should consider shifting the focus to two or three timepoints only.

Secondly, the authors suggest a significantly faster decrease in YBOCS scores of the am-GPi group compared to the Voi-Cm/PF group (see line 200, line 291, line 323). However, after the reviewer, the significant difference in baseline-YBOCS scores between the groups, which is likely to drive the mentioned difference in decreasing slope (see Figure 4), significantly hampers the validity of this message. Maybe the authors would need to think about a relative measure of change and use this to run the statistics. If not, please provide information as to why the difference of baseline-conditions does not interfere in this type of analysis.

Thirdly, the reviewer assumes that the random intercept model was superior to a random intercept and random slope model (line 115)? Furthermore: They have used a linear regression model (line 167). Looking at the data I wonder if a non-linear model would not be superior in fitting the data? Finally: in line 168 the authors calculate the average decrease between two timepoints. I question the usefulness of this approach, given the non-linear decrease in scores.

Minor concerns:

Line 19: DBS in the ventralis oralis/centromedian-parafascicular nucleus of the thalamus…

Line 33: Keywords: instead of limbic circuits, list the am-GPi and Voi-Cm/PF

Line 42: …and other limbic structures…

Line 43: … 1% and the onset…

Line 59: OCB/OCD…

Line 101/105: Figure 1 & 2: please add information on field strength and provide the reader with information such as whether you provide T1 or T2 images…

Lines 121-124: Thank you for providing this information. For the paper, I think this can be omitted (in particular, the comment about multicollinearity, which the authors point out correctly).

Line 134: Maybe write: in compliance with the Declaration of Helsinki and its later...

Line 148: Please add the follow-up timepoints also in the table caption.

Line 154: Please add the follow-up timepoints also in the table caption.

Line 172: Please include t0-t4 in the tables, in order to make it more reader-friendly.

Line 186: The statistical method used is missing.

Line 188: Please indicate p=0.13

Line 195 & 201: Refer to colour-code: slopes of am-GPi (red) etc

Line 211: 13 patients underwent electrode removal: when? Are these patients partially included in the sample?

Line 212: Please change to (n=8), (n=3) and (n=2), respectively.

Line 215: Inappropriate phrasing of the sentence. Please change.

Line 221: Exchange to “and” instead of “,”.

Line 239: Change “target” to “targets”.

Line 239/240: The authors write: ”Currently, targets are selected based on the specific clinical presentation in a personalized and tailored approach.” However, in the introduction, the authors rightly outline the scarcity of studies investigating efficacy of different targets for the treatment of GTS using adequate samples and research methods. This very fact goes somewhat against their claim about tailored interventions. Please relativize.

Line 244: The reviewer is not sure whether Figure 5 adds important information to the paper, apart from the sample sizes at each follow up timepoint, which should anyways be added to the text and maybe the tables. If the authors want to keep Figure 5, it should be repositioned accordingly. Here, the authors discuss the selection of different targets based on clinical criteria.

Line 247: Please add: … parafasciular nucleus of the thalamus …

Line 255: Please add: … was 50.5% in the anterior…

Line 273: Please change “studies” to “study”.

Line 286: Please get rid of the p-value as this is normally not reiterated in the discussion section.

Line 304: At the very least, the fact of a statistical difference of group on YBOCS at baseline and its possible implications should be mentioned in the limitations.

Line 322: The authors write: “a brilliant response to DBS”, please rephrase.

Line 331: Author Contributions is missing.

Acknowledgment for English proof reading could be added.

Author Response

  1. The authors refrain from outlining the sample size of patients at each different follow-up timepoint. According to Figure 5 (line 245), the reviewer assumes 5 patients at the maximum that completed the long-term follow up in the am-GPi study arm. If that were the case, the low sample size would not only impact on the statistical analysis but also on the confidence with which one can draw conclusions. Underpowered studies are a XX. Therefore, please provide the exact sample size for each study-group (Voi-Cm/PF and am-GPi) and at each follow-up. Please provide this important bit of information (lines 138 and following) so that the reviewer can assess the validity of the results and make sure, that the study is not underpowered (causing the danger of inflating effects where there are none). Otherwise, the authors would need to emphasize that parts of their analysis were performed using a small sample, outlining why this was justifiable from a statistics point of view and highlight that caution is warranted when interpreting these results. Maybe they should consider shifting the focus to two or three timepoints only.

As asked by the reviewer, we attach the full descriptive statistics of the present study. Please see lines 127 and 132 and the new tables attached.

  1. Secondly, the authors suggest a significantly faster decrease in YBOCS scores of the am-GPi group compared to the Voi-Cm/PF group (see line 200, line 291, line 323). However, after the reviewer, the significant difference in baseline-YBOCS scores between the groups, which is likely to drive the mentioned difference in decreasing slope (see Figure 4), significantly hampers the validity of this message. Maybe the authors would need to think about a relative measure of change and use this to run the statistics. If not, please provide information as to why the difference of baseline-conditions does not interfere in this type of analysis.

We acknowledge the reviewer's concerns regarding patient's different YBOCS baseline score, which renders comparison between the two groups less intuitive and more challenging to analyze. Unfortunately our study was not a randomized trial and therefore has its own limitations. That said, with a correct interpretation of the results, we are convinced that important insights can nonetheless be drawn. We agree that our findings don't necessarily imply that amGPi treatment leads to a better long-term (and consequently overall) OCD symptom relief. What the paper demonstrates is that in the observed period of time (48 months follow-up) the  OCD symptom reduction rate (measured by the YBOCS score difference from baseline to 48 month's follow up) was considerably higher in the amGPi group than in the thalamus'. It is important to emphasize this point because two hypothesis can therefore be formulated: 1) the amGPi target leads both to a faster and quantitatively higher OCD symptom relief 2) amGPi target only has an initial higher decrease rate, but it is possible that in the long term this trend reaches "a plateau" by which at the end its overall efficacy results comparable to the one of the thalamus. We are not excluding the second hypothesis, but at the same time we think that it's also legitimate to consider the former. Future prospective studies conceived to test whether one target is superior to the other in the treatment of obsessive-compulsive disorder's comorbidity will be needed in order to shed light on the problem.

  1. Thirdly, the reviewer assumes that the random intercept model was superior to a random intercept and random slope model (line 115)? Furthermore: They have used a linear regression model (line 167). Looking at the data I wonder if a non-linear model would not be superior in fitting the data? Finally: in line 168 the authors calculate the average decrease between two timepoints. I question the usefulness of this approach, given the non-linear decrease in scores.

We favored a random intercept model over a random intercept and random slope one based on their significantly lower scores with regards to Akaike's Information Criterion (AIC) of 1920.98 vs 2053.77 (for YGTSS) and 1515.84 vs 1593.97 (for YBOCS). Curvilinear model could have possibly been a better individual fit for the distributions, yet it renders confrontation between the two slopes (which was the major focus of our study) problematic; for this reason we decided to opt for a linear regression model which would allow simple comparison, providing nonetheless an acceptable fit for our score distributions.

We agree with you when considering the computation of the average decrease between two time points simply a model (with all its limitations), but we think it could be useful to give the reader an idea of the general efficacy throughout time of the two different surgical treatments on patient's symptoms severity reduction.

We thank the reviewer for his/her precious comments and suggestions. All the minor concerns has been revised as requested. All the changes in the manuscript has been highlighted using the “Track-Changes” function in Microsoft Word